# Nanoimaging of resonating hyperbolic polaritons in linear boron nitride antennas

F.J. Alfaro-Mozaz[1], P. Alonso-González[1,2], S. Vélez[1], I. Dolado[1], M. Autore[1], S. Mastel[1], F. Casanova[1,3], L.E. Hueso[1,3], P. Li[1], A.Y. Nikitin[1,3] & R. Hillenbrand[3,4]

Polaritons in layered materials—including van der Waals materials—exhibit hyperbolic dispersion and strong field confinement, which makes them highly attractive for applications including optical nanofocusing, sensing and control of spontaneous emission. Here we report a near-field study of polaritonic Fabry–Perot resonances in linear antennas made of a hyperbolic material. Specifically, we study hyperbolic phonon–polaritons in rectangular waveguide antennas made of hexagonal boron nitride (h-BN, a prototypical van der Waals crystal). Infrared nanospectroscopy and nanoimaging experiments reveal sharp resonances with large quality factors around 100, exhibiting atypical modal near-field patterns that have no analogue in conventional linear antennas. By performing a detailed mode analysis, we can assign the antenna resonances to a single waveguide mode originating from the hybridization of hyperbolic surface phonon–polaritons (Dyakonov polaritons) that propagate along the edges of the h-BN waveguide. Our work establishes the basis for the understanding and design of linear waveguides, resonators, sensors and metasurface elements based on hyperbolic materials and metamaterials.

[1] CIC nanoGUNE, 20018 Donostia-San Sebastián, Spain. [2] Departamento de Física, Universidad de Oviedo, 33007 Oviedo, Spain. [3] IKERBASQUE, Basque Foundation for Science, 48013 Bilbao, Spain. [4] CIC nanoGUNE and EHU/UPV, 20018 Donostia-San Sebastian, Spain. Correspondence and requests for materials should be addressed to A.Y.N. (email: a.nikitin@nanogune.eu) or to R.H. (email: r.hillenbrand@nanogune.eu).

Polaritons—quasiparticles resulting from the coupling of photons and oscillating charges[1]—in nanostructured materials enable strong confinement and enhancement of electromagnetic fields at extreme subwavelength-scale dimensions[2]. For that reason, they have numerous applications, for example in single molecule sensing[3], heat-assisted magnetic recording[4] and photocatalysis[5]. Typically, propagating and localized surface plasmon–polaritons—collective oscillations of free electrons at metal or semiconductor surfaces coupled to electromagnetic fields[6–8]—have been employed. A promising alternative, which is by far less considered yet, are surface phonon–polaritons in polar crystals[9–11]. These quasiparticles—resulting from the coupling of electromagnetic fields and crystal lattice vibrations—exist from mid-infrared to THz frequencies in the so-called reststrahlen band (defined as the region between the transversal and longitudinal optical phonon frequencies, TO and LO, respectively), where the real part of the crystals' dielectric permittivity is negative. They offer significantly improved field enhancements and quality factors compared to plasmons[9,10]. At mid-infrared frequencies, they have been studied, for example, in SiC[10] and quartz[11], and could find applications in thermal emission control[12–14] and sensing[15,16].

The emergence of van der Waals materials opens novel opportunities for polariton-based photonic technologies[17–19]. Particularly, hexagonal boron nitride (h-BN) exhibit mid-infrared phonon–polaritons with remarkably low losses[20–22], while high-quality single-crystalline layers are easy to prepare by exfoliation. Furthermore, due to its layered crystal structure (uniaxial anisotropy), the permittivity tensor $\epsilon$ is diagonal, with $\epsilon_{zz} = \epsilon_{\parallel}$ and $\epsilon_{xx} = \epsilon_{yy} = \epsilon_{\perp}$ being the components parallel and perpendicular to the anisotropy axis, respectively. When $\mathrm{Re}(\epsilon_{\parallel}) \cdot \mathrm{Re}(\epsilon_{\perp}) < 0$, the phonon–polaritons propagate inside the material and exhibit a hyperbolic dispersion[23,24], that is, the isofrequency surface of the polariton wavevector $\mathbf{q}(\omega) = (q_x, q_y, q_z)$ presents a hyperboloid. For h-BN, we find two reststrahlen bands, where one of the permittivity components is negative. In the lower reststrahlen band (760–825 cm$^{-1}$), the real part of the out-of-plane permittivity $\mathrm{Re}(\epsilon_{\parallel})$ is negative (Type I hyperbolic dispersion), while in the upper reststrahlen band (1,360–1,614 cm$^{-1}$), the real part of the in-plane permittivity $\mathrm{Re}(\epsilon_{\perp})$ is negative (Type II hyperbolic dispersion). The hyperbolic polaritons[22,26] can propagate with negative phase velocity and slow group velocity[24], showing extreme mode confinement in thin slabs[22,23,25] and nanotubes[26]. Large quality factors of phonon–polariton resonances (up to 283) in h-BN nanocones have been recently reported in ref. 21, promising that phononic antennas made of hyperbolic materials have strong application potential for infrared nanophotonics. Surprisingly, linear waveguide antennas—the most canonical type of antennas, and basic building block in many nanophotonic devices—made of h-BN have not been investigated yet.

Here we study experimentally and theoretically linear antennas (rods with nanoscale cross-section) made of the van der Waals material h-BN. We perform infrared nanospectroscopy and nanoimaging of the antennas in the upper reststrahlen band, yielding the antennas' local spectral near-field response and modal near-field distribution, respectively. We find resonances of high-quality factors of about 100 and puzzling near-field patterns. By full-wave simulations we perform a comprehensive analysis of both the experimental images and near-field spectra. We show that the resonances and exotic near-field patterns are governed, interestingly, by hyperbolic surface phonon–polaritons (Dyakonov polaritons[27–31]) rather than volume phonon–polaritons.

## Results

**s-SNOM and nano-FTIR set-up.** For infrared nanoimaging we used a scattering-type scanning near-field optical microscope (s-SNOM[32–35]). It is equipped with a Fourier transform spectrometer, which allows for nanoscale Fourier transform infrared (nano-FTIR[36]) spectroscopy of the antennas. The nanoimaging and nanospectroscopy concept is illustrated in Fig. 1a. The metallic scanning probe tip (cantilevered standard Pt-coated silicon tip) of the s-SNOM is illuminated from the side with a p-polarized infrared beam of electric field $E_{\mathrm{inc.}}$ Acting as an infrared antenna, the tip concentrates the incident field into a nanoscale spot at the apex. This nanoscale 'hot spot' acts as a local source for launching hyperbolic phonon–polaritons (HPs) in the h-BN antennas. The HPs are reflected at the ends of the antennas, giving rise to Fabry–Perot resonances. The strongly enhanced near fields caused by these resonances are imaged and spectrally analysed by recording the field scattered by the tip, $E_s$.

For nanoimaging, we illuminated the tip with the monochromatic infrared light of a frequency-tunable quantum cascade laser (QCL) and recorded the amplitude of the tip-scattered field with a pseudo-heterodyne Michelson interferometer[34] as a function of tip position (right side in Fig. 1b, coloured in red), yielding near-field images (Fig. 1d). For nano-FTIR spectroscopy[36], the broadband infrared radiation from a laser supercontinuum was used for illuminating the tip. The tip-scattered light, $E_s$, was recorded with an asymmetric Fourier transform spectrometer (left side in Fig. 1b, coloured in blue). The recorded spectra were normalized to a gold reference, $E_{s,\mathrm{Au}}$, yielding near-field amplitude spectra, $|E_s/E_{s,\mathrm{Au}}|$, at a fixed tip position (point spectroscopy) (Fig. 1c). By recording point spectra as a function of the tip position, we obtained high-resolution spectral line scans.

**Nanospectroscopic analysis of a linear h-BN antenna.** We first perform nano-FTIR spectroscopy (Fig. 2a) of a representative, 1,723 nm long h-BN rod antenna (see Supplementary Fig. 1) at its extremity and centre (positions marked by red and blue dots, respectively, in the topography image shown in Fig. 2c). Within the upper reststrahlen band we observe several spectrally sharp and closely spaced peaks (denoted by $n = 1, \ldots, 6$ in Fig. 2a) at frequencies $\omega_n$, indicating antenna resonances. From the width of the peaks, we estimate quality factors ($Q$) in the range of $Q_n \sim 80$–120 (see 'Methods' section). They are comparable to that of h-BN nanocones[21] and SiC nanopillars[8], and higher than the $Q$-factors reported for metal (plasmonic) antennas in the mid-infrared spectral range[37]. The corresponding lifetimes of the resonances ($\tau_n = 2Q_n/\omega_n$) are in the order of half a picosecond.

To identify the origin of the resonances, we performed a spectral line scan (Fig. 2d) along the axis of the antenna (marked by dashed line in Fig. 2c). For each peak $n$, we find strong near-field signal oscillations along the antenna axis. The number of near-field signal maxima and minima is steadily growing with increasing frequency, while the distance between them (along the antenna axis) decreases. Such behaviour is a clear indication of a spectrally evolving standing wave pattern, that is, longitudinal Fabry–Perot resonances in the h-BN rod, similar to plasmon resonances in metal rods[38,39]. We can clearly distinguish the first six longitudinal resonances in Fig. 2d, corresponding to the $n = 1, \ldots, 6$ peaks in Fig. 2a. The first-order dipolar resonance ($n = 1$) manifests at 1,370 cm$^{-1}$ by the two near-field signal maxima close to the rod extremities, while the second-order ($n = 2$) resonance (1,395 cm$^{-1}$) exhibits a strong near-field signal in the centre of the rod and two less pronounced near-field signal maxima at the rod extremities. The third-order ($n = 3$) resonance at 1,411 cm$^{-1}$ clearly shows the four typical near-field signal maxima along the antenna axis. The trend continues up to the

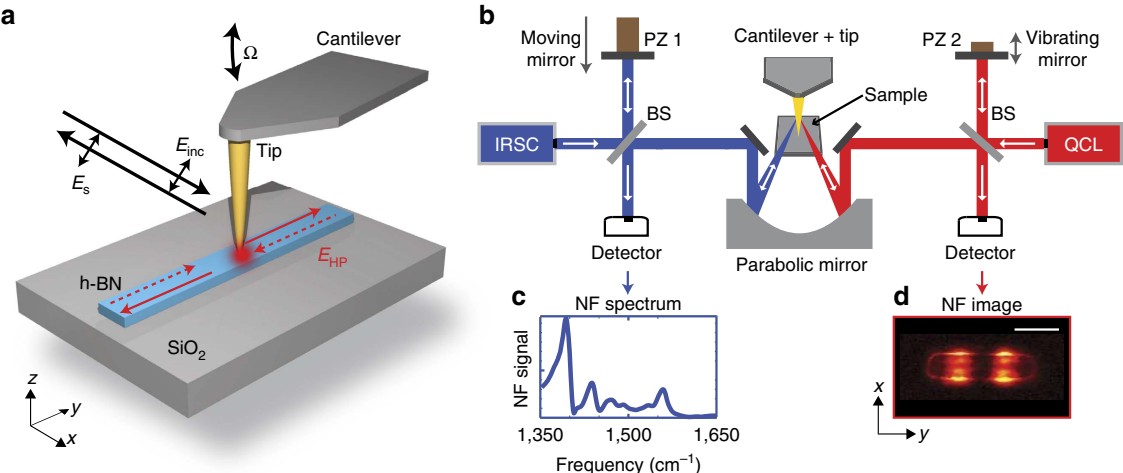

**Figure 1 | Infrared nanoimaging and nanospectroscopy of linear h-BN antennas.** (**a**) Illustration of a metallic cantilevered tip probing a linear antenna of h-BN. The tip oscillates with frequency $\Omega$ to allow complete background suppression (See 'Methods' section). (**b**) Schematics of the s-SNOM/nano-FTIR set-up. For nano-FTIR spectroscopy, we illuminate the tip with a broadband laser (infrared supercontinuum, IRSC). The backscattered light is analysed with an asymmetric Fourier transform spectrometer (left). BS, beamsplitter; PZ1, piezo-actuated moving mirror. For s-SNOM nanoimaging, we illuminate the tip with a frequency-tunable QCL. The backscattered light is detected with a Michelson interferometer (right). PZ 2, piezo-actuated vibrating mirror. (**c,d**) Examples of the spectra and images taken in this work, respectively. Scale bar, 500 nm (**d**).

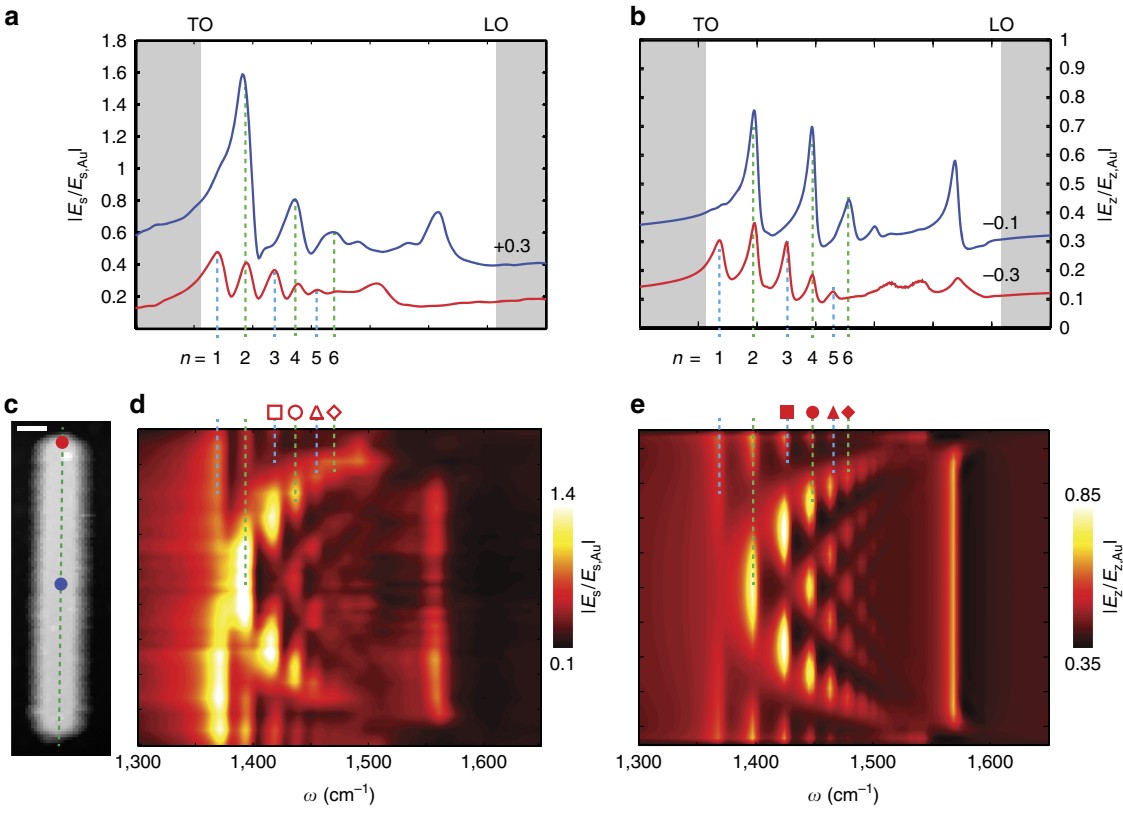

**Figure 2 | Nano-FTIR study of a 1,723 nm long linear h-BN antenna on SiO₂.** (**a**) Experimental and (**b**) simulated nano-FTIR amplitude spectra. They were obtained at the two positions marked by the red and blue dots in **c**, and normalized to the nano-FTIR spectrum of a gold film. For clarity, the spectra have been vertically shifted by the indicated values. The white background colour indicates the reststrahlen band. (**c**) AFM topography image of the antenna. The antenna has a thickness of 74 nm (measured from the AFM image). Its average width and height were measured by electron microscopy and amount to 190 nm and 1,723 nm, respectively (see Supplementary Fig. 1). Scale bar, 200 nm. (**d**) Experimental and (**e**) simulated spectral line scan along the dashed green line in **c**. Open and solid symbols mark the resonances analysed in Fig. 3b.

sixth order. However, the peaks become less pronounced, which we attribute to the increase of losses for higher-order longitudinal antenna modes (analogous to plasmonic antennas[38]).

**Numerical model.** We corroborated our experimental data with full-wave simulations of the near-field spectra. To that end, we used a model where the metallic tip is approximated by a dipole

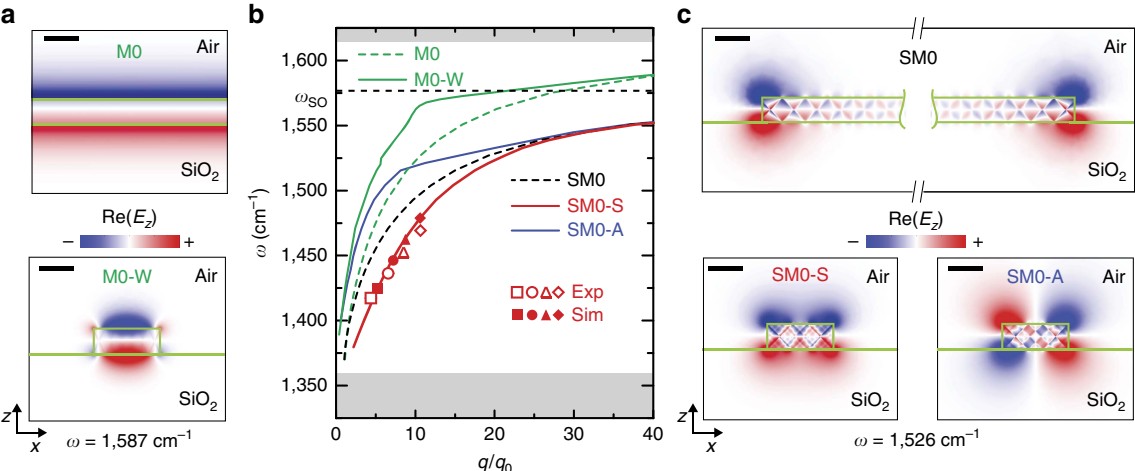

**Figure 3 | Dispersion and near-field distribution of hyperbolic phonon–polaritons in h-BN. (a)** Near-field distribution of volume modes of a slab (M0), and of a rectangular waveguide (M0-W). **(b)** Dispersion curves of volume modes (green solid and dashed lines) and surface modes (black dashed line (SM0), and red and blue solid lines (SM0-S and SM0-A)). The experimental (open symbols) and calculated (solid symbols) data points ($q_n$, $\omega_n$) were extracted from Fig. 2d,e, respectively. The different symbols represent the data obtained from the different resonance orders $n$. The horizontal axis is normalized to the momentum of light in free space, $q_0$. The white area marks the reststrahlen band. The horizontal black dashed line indicates the asymptote for surface modes ($\omega_{SO}$). **(c)** Near-field distribution of the fundamental surface mode of a semi-infinite slab (SM0), the symmetric surface mode (SM0-S) and the antisymmetric surface mode (SM0-A) of a rectangular waveguide. The width and height of the rectangular waveguide are 190 nm and 74 nm, respectively. The height of the slab is 74 nm. Scale bars, 100 nm **(a,c)**.

source located above the antenna (see 'Methods' section). This model has been shown to reliably reproduce the experimental near-field images in graphene nanoresonators[35]. By plotting the normalized near field below the dipole as a function of frequency $\omega$ (for a fixed dipole position), we obtain near-field spectra, $|E_s/E_{s,Au}|$. By plotting the near-field spectra as a function of the lateral dipole position, we obtain spectral near-field line scans. The simulated point spectra at the rod extremity and centre are shown in Fig. 2b (red and blue points in Fig. 2c, respectively), while the spectral line scan along the antenna axis is displayed in Fig. 2e. We find an excellent qualitative agreement with the experimental data of Fig. 2a,d. Our model can be thus applied for a combined experimental and theoretical in-depth analysis of Fabry–Perot resonances in linear h-BN antennas.

**Dispersion and analysis of the resonating mode.** For studying the dispersion $\omega(q)$ of the HP mode yielding the longitudinal Fabry–Perot resonances, we extracted from both the experimental (Fig. 2d) and simulated (Fig. 2e) spectral line scans (see Supplementary Fig. 2) the HP wavevector $q_n$ at the resonance frequency $\omega_n$ of the longitudinal $n$-th order mode. The wavevectors $q_n$ are parallel to the longitudinal axis of the antenna ($y$ axis) and to the atomic layers of the h-BN slab out of which the antennas were fabricated. In good approximation, they can be calculated by using the relation $q_n = \pi/L_n$, where $L_n$ is the distance between the adjacent maxima in the central part of the antenna (for each resonance order $n > 2$). To be more precise, we determined the $q_n$ by fitting the line profiles with a sine function, as described in the Supplementary Fig. 2. In Fig. 3b, we show the dispersion obtained from experiment (filled symbols) and simulation (open symbols). We observe that the wavevector $q$ is significantly increased compared to that of free-space photons of the same energy, $q_0$, and strongly increases with frequency. Both findings indicate that highly confined waveguide polaritons are the cause of the longitudinal Fabry–Perot resonances. To identify the waveguide mode, we performed a numerical analysis of the electromagnetic modes in an infinitely long h-BN waveguide, which has a cross-section that is similar to that of the antenna

(see 'Methods' section). As expected for hyperbolic material waveguides[40], we found a variety of coexisting waveguide modes. The mode with the smallest wavevectors (mode profile M0-W in Fig. 3a; green solid line in Fig. 3b) can be identified as the fundamental volume-confined HP mode. Indeed, at large wavevectors its dispersion curve converges to that of volume-confined HPs in an infinite h-BN slab of the same thickness (mode profile M0 in Fig. 3a, green dashed line in Fig. 3b). Because of the strong discrepancy of the M0-W dispersion curve with the data points (symbols), we can exclude that this mode is observed in the h-BN antennas. Our mode analysis also reveals modes with larger wavevectors and symmetric and antisymmetric spatial distribution of the vertical component of the electric field, $Re(E_z)$ (mode profiles SM0-S and the SM0-A in Fig. 3c; solid red and blue dispersion curves in Fig. 3b). Interestingly, the dispersion curve of the SM0-S mode matches perfectly the measured data points (symbols), which let us conclude that this mode is the root cause of the observed Fabry–Perot resonances in the h-BN antennas.

To elucidate the physical nature of the SM0-S mode, we calculated its dispersion $\omega(q)$ up to wavevectors $q \approx 120\ q_0$. We find that $\omega(q)$ asymptotically approaches the frequency $\omega_{SO} = 1{,}576\ \text{cm}^{-1}$ (horizontal black dashed line in Fig. 3b). Interestingly at $\omega_{SO}$, the condition $\epsilon_\perp = -1$ is fulfilled, which typically determines the dispersion limit of surface polaritons[1]. We thus conclude that the SM0-S mode is a hyperbolic surface polariton mode, rather than a hyperbolic volume polariton mode. Indeed, a uniaxial (layered) material can support surface waves (also called Dyakonov surface waves[41,42]) at the surfaces that are perpendicular to its layers. Particularly, when the material exhibits hyperbolic dispersion, the supported modes are called hyperbolic surface polaritons (or Dyakonov polaritons[28–30,43]). This condition is fulfilled at the edges of h-BN slabs. Because of the finite thickness of the slabs, however, the hyperbolic surface polaritons propagate along the edges as guided modes, as we confirmed by s-SNOM imaging of the edges of large h-BN flakes[44]. The dispersion of the lowest guided surface mode in a thin semi-infinite h-BN slab (mode profiles SM0 in Fig. 3c) is shown by the dashed black line in Fig. 3b. It lies between the

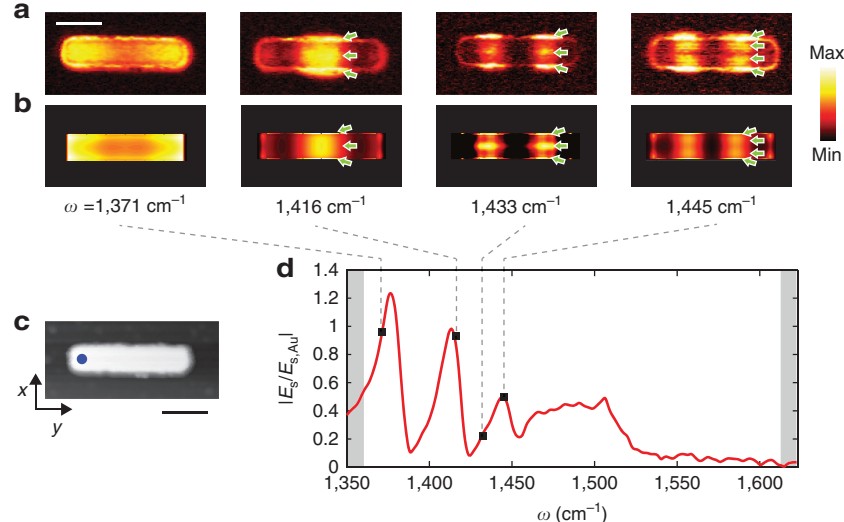

**Figure 4 | Mode mapping of a linear h-BN antenna of 1,220 nm in length.** (**a**) Experimental s-SNOM images recorded at frequencies marked by black squares in **d**. Scale bar, 0.4 μm. (**b**) Simulated s-SNOM images. Green arrows in **a,b** indicate the position of the near-field maxima along the transverse direction. (**c**) AFM topography image of the antenna. Scale bar, 0.4 μm. (**d**) Nano-FTIR spectrum recorded at the position marked by the blue dot in **c**. The white background colour indicates the reststrahlen band. The height of the antenna measured in the AFM was 65 nm, and the average width obtained from the AFM topography was 250 nm.

calculated SM0-S and SM0-A modes of the h-BN waveguide, while sharing the same asymptote at $1,576 \, cm^{-1}$ (horizontal black dashed line in Fig. 3b). We conclude that the SM0-S and SM0-A modes of the h-BN waveguide are hybridized surface modes (exhibiting symmetric and antisymmetric field distributions, respectively), which result from the electromagnetic coupling of the SM0 mode propagating at the two opposing edges of the h-BN waveguide (see Fig. 3c). Comparing the spectral mode analysis (red data points in Fig. 3b) with the various calculated mode dispersions (solid and dashed curves in Fig. 3b), we can finally conclude that our h-BN antennas exhibit Fabry–Perot resonances of the SM0-S mode.

We note that the near fields at the tip apex, in principle, can excite any of the modes found in Fig. 3. However, in the spectral line scans of Fig. 2d,e, we only observe the SM0-S mode. We explain this finding by the strong damping of the M0-W mode, which propagates less than one polariton wavelength (see Supplementary Fig. 3). This mode might be observed in the future with wider and thinner antennas, where the damping of the mode can be significantly reduced (see Supplementary Figs 4 and 5). On the other hand, the SM0-A mode cannot be excited when the tip is scanned along the longitudinal antenna axis, where the field of the SM0-A mode is zero. This mode, however, might be excited and observed in the future by scanning the tip, for example, along one of the long antenna edges.

**Near-field imaging of h-BN antennas.** We gain further insights into the Fabry–Perot resonances by studying both near-field spectrum and near-field images of a shorter h-BN rod antenna (Fig. 4). The topography of the 1,220 nm long rod is shown in Fig. 4c. The near-field spectrum (recorded at the position marked by a dot in Fig. 4c) again reveals several peaks (Fig. 4d). They correspond to the different longitudinal resonance orders, which are readily identified with the help of the experimental (Fig. 4a) and simulated (Fig. 4b, matching well the experiment) near-field images. With increasing frequency, we clearly observe an increasing number $n$ of near-field signal maxima along the antenna, corresponding to the $(n-1)$-th resonance order (note that the maxima at the rod extremities are weak, similar to

electron energy loss spectroscopy images[39,45]). Surprisingly, we also observe near-field signal oscillations in the transversal direction (marked by green arrows in Fig. 4a,b), which have no analogue in conventional plasmonic antennas. Their periodicity is much shorter than the wavelength of the SM0-S mode, which lets us exclude transverse polariton resonances. We further note that the number of transverse near-field signal maxima does not match the number of near-field signal maxima in longitudinal direction. For example, the third-order longitudinal antenna resonance exhibits both three and four transverse near-field signal maxima at $1,433 \, cm^{-1}$ and $1,445 \, cm^{-1}$, respectively.

In Fig. 5 we elucidate the origin of the transverse structure of the antenna mode by numerically analysing the SM0-S mode of the corresponding h-BN waveguide (dispersion shown in Fig. 5b). To that end, we plot the calculated transverse near-field profiles (5 nm above the surface of the waveguide) at different frequencies (Fig. 5c) and compare them with experimental s-SNOM profiles (Fig. 5d, extracted from near-field images such as the ones shown in Fig. 4a). An excellent agreement is observed. Particularly, we find that the number of transverse near-field signal maxima does not monotonously increase with increasing frequency. We clarify this observation by plotting in Fig. 5a the simulated near-field profile of the SM0-S mode, $Re(E_z)$, at two different frequencies. Inside the h-BN waveguide, we recognize 'zig-zag' patterns. They manifest HP rays that emerge from corners[21,46,47] and reflect at the top and bottom h-BN waveguide surfaces[48]. As the propagation angle, $\theta$, of the hyperbolic rays depends on the frequency $\omega$ (given by[21] $\tan \theta(\omega) = i\sqrt{(\epsilon_\perp(\omega))}/\sqrt{\epsilon_\parallel(\omega)}$), the multiple reflections yield complex and frequency-dependent field distributions inside the waveguide, which extend several nanometres above the top waveguide surface. Scanning the tip in close proximity across the top waveguide surface (perpendicular to the longitudinal waveguide axis) thus leads to the transversal near-field oscillations observed in Fig. 5c (marked by green arrows). On the other side, we observe in Fig. 5a a rather homogeneous near-field distribution at distances larger than 100 nm both above the top waveguide surface and below the bottom surface. Its structure (opposite sign of the vertical component of the electric field above and below the waveguide) does not change with frequency, which reveals that the different

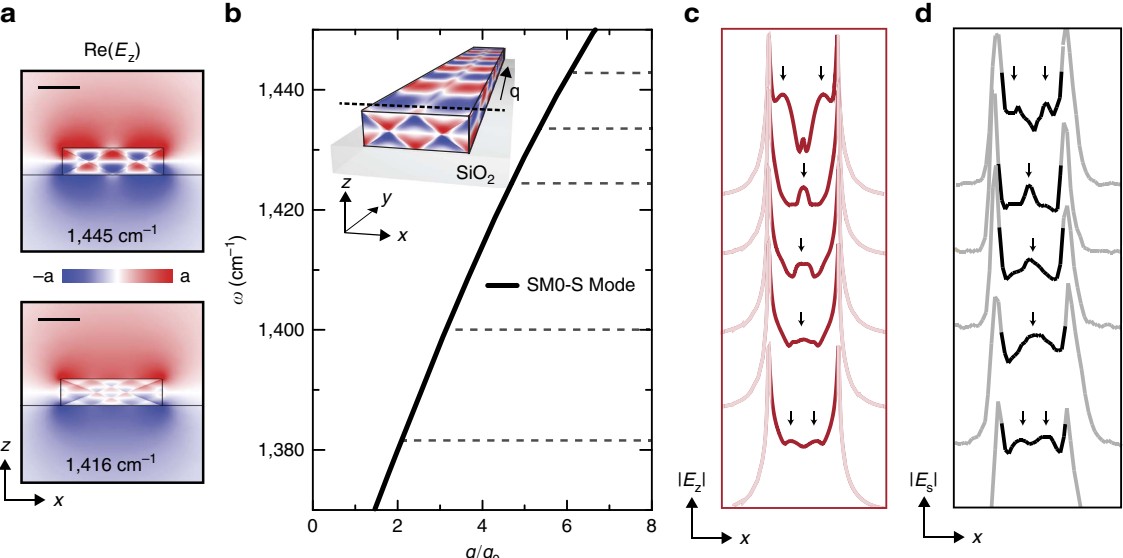

**Figure 5 | Analysis of the SM0-S mode of a rectangular waveguide of 250 nm width and 65 nm thickness.** (**a**) Simulated near-field distribution (Re($E_z$)) of the SM0-S mode at 1,445 cm$^{-1}$ (top) and 1,416 cm$^{-1}$ (bottom). Scale bar, 100 nm (**a**). (**b**) Calculated dispersion of the SM0-S mode. The inset shows the schematics of the waveguide and the near-field distribution of the SM0-S mode at 1,445 cm$^{-1}$. The dashed line indicates the direction of the near-field profiles in **c,d**. (**c**) Calculated near-field profile of $|E_z|$ in a height of 5 nm above the waveguide surface of the waveguide at 1,445, 1,432, 1,424, 1,400 and 1,381 cm$^{-1}$ (from top to bottom) (**d**) Experimental near-field profiles measured at 1,443, 1,432, 1,424, 1,416 and 1,408 cm$^{-1}$ (from top to bottom). For better visibility of the near-field variations, we plot the central part of the near-field profiles in **b,c** in darker colour. For a better comparison of calculated and experimental near-field profiles in **c,d**, we mark local near-field maxima by black arrows.

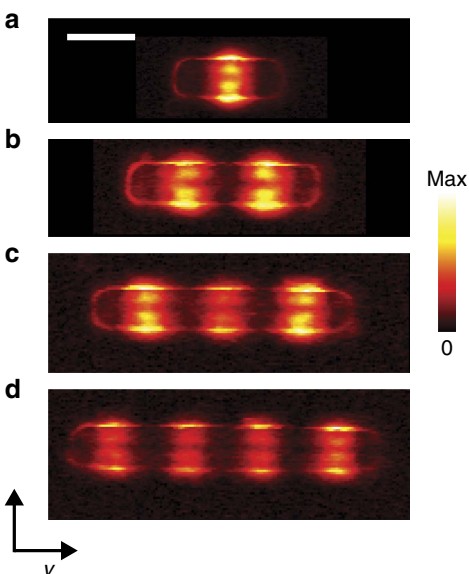

**Figure 6 | s-SNOM images of a set of linear h-BN antennas of different length $L$.** Imaging frequency $\omega = 1,432$ cm$^{-1}$. The width and thickness of each antenna (according to the AFM topography images) are 230 nm and 64 nm, respectively. (**a**) $L = 746$ nm. (**b**) $L = 1,327$ nm. (**c**) $L = 1,713$ nm. (**d**) $L = 2,210$ nm. Scale bar, 500 nm.

near-field profiles shown in Fig. 5c,d belong to the same waveguide mode. Strikingly, our results reveal that the transverse near-field structure of one specific hyperbolic waveguide mode (here the SM0-S mode) can exhibit significant variations on frequency change.

Finally, we performed near-field imaging of differently long antennas of the same width and height (Fig. 6). With increasing

antenna length $L$, the images (all recorded at 1,432 cm$^{-1}$) exhibit an increasing number of near-field signal maxima along the antenna, revealing the different resonance orders. Importantly, the transverse near-field profile is identical for all antennas (up to 2,210 nm in length), confirming that the exotic transverse near-field profile of the SM0-S mode is a robust and intrinsic feature of the waveguide mode, and not affected by the antenna length.

## Discussion

In conclusion, we employed real-space infrared nanoimaging and nanospectroscopy for studying Fabry–Perot resonances of hyperbolic phonon–polaritons in finite-length h-BN waveguides acting as infrared antennas. We found resonances exhibiting $Q$-factors of about 100, which make the antennas interesting building blocks for the development of infrared molecular sensors, narrowband thermal emitters or metasurfaces for flat infrared photonic elements. By a detailed mode analysis, we demonstrated that the waveguide mode exhibiting longitudinal Fabry–Perot resonances originates from the hybridization of hyperbolic surface phonon–polaritons (Dyakonov polaritons) that propagate along the edges of the h-BN antennas (respectively, the h-BN waveguides). This mode exhibits a stronger field confinement (that is, larger wavevector) compared to the waveguides' volume modes (of the same order). We note that the odd longitudinal Fabry–Perot resonances can be excited by far-field illumination, which could enable, for example, field-enhanced far-field spectroscopy applications. From a general perspective, our results provide valuable insights into the fundamental properties of polariton modes in deep subwavelength-scale linear waveguides based on naturally or artificially layered materials, such as van der Waals materials and heterostructures or metal-dielectric metamaterials. The knowledge about the mode properties will be of critical importance for the development of photonic circuits based on

hyperbolic plasmon–polariton and phonon–polaritons in linear waveguides and antennas.

## Methods

**Fabrication of h-BN antennas.** Large and homogeneous h-BN flakes were isolated on a Si/SiO2 (250 nm) substrate. To that end, we first performed mechanical exfoliation of commercially available h-BN crystals (HQ graphene Co, N2A1) using blue Nitto tape (Nitto Denko Co., SPV 224P). Then, we performed a second exfoliation of the h-BN flakes from the tape onto a transparent polydimethyl-siloxane stamp. After that, via both optical inspection and atomic force microscope (AFM) characterization of h-BN flakes on the stamp, high-quality flakes with large areas and required thickness were identified and transferred onto a Si/SiO2 (250 nm) substrate using the deterministic dry transfer technique[49,50].

The h-BN nanoantennas were fabricated from the h-BN flakes via high-resolution electron beam lithography using a double-layer poly(methyl methacrylate) (PMMA) resist (495 A4/950 A2), e-beam evaporation of Cr (3 nm) and thermal evaporation of Al (40 nm) onto the developed resist, lift-off in acetone, and chemical etching of the h-BN flake with a SF6/Ar 1:1 plasma mixture at 20 sccm flow, 100 mTorr pressure and 100 W power for 30 s (RIE OXFORD PLASMALAB 80 PLUS reactive ion etcher). Finally, the metal mask was removed by immersing the sample in chromium etchant (Sigma-Aldrich Co., 651826) for 20 min, rinsed in deionized water and dried using a $N_2$ gun.

**Near-field imaging.** The scattering-type SNOM used for this work (Neaspec GmbH, Germany) is based on an AFM. The vertically oscillating conventional metal-coated (Pt/Ir) tip (ARROW-NCPt-50, Nanoworld) acts as near-field probe. The tapping frequency and amplitude was $\Omega \approx 270$ KHz and about 100 nm, respectively. Tip and sample were illuminated with the p-polarized mid-infrared laser beam of a QCL (tunable $1,295–1,445$ cm$^{-1}$, Daylight Solutions, USA), focused with a parabolic mirror (NA = 0.4), at an angle of 60° relative to the surface normal. The p-component of the tip-scattered light $E_S$, collected with the same parabolic mirror, was recorded with a pseudo-heterodyne Michelson interferometer (right side of Fig. 1b), allowing for detection of both scattering amplitude and phase signals[34]. To suppress the background scattering from the tip shaft and sample, the detector signal was demodulated at a frequency $4\Omega$. By 2D scanning the sample below the tip, an infrared near-field image is recorded.

**Nano-FTIR.** For nano-FTIR spectroscopy, the same basic s-SNOM set-up (AFM + parabolic mirror) described above is used. Tip and sample are illuminated with a broadband mid-infrared laser supercontinuum (spanning $1,200–1,700$ cm$^{-1}$, average power of 1 mW), which is generated by difference frequency generation using a Femtofiber pro IR and a SCIR laser from Toptica (Germany). Fourier transform spectroscopy is accomplished as follows. The tip-scattered signal is analysed with an asymmetric Fourier transform spectrometer (left side in Fig. 1b), which is based on a Michelson interferometer. In contrast to conventional FTIR spectroscopy techniques, tip and sample are located in one of the interferometer arms[36]. The detector signal is demodulated at a frequency $4\Omega$ for effective background suppression. An interferogram is measured by recording the demodulated detector signal as a function of the position of the reference mirror, $d$, at a fixed tip position. Subsequent Fourier transform of the recorded interferogram yields the complex-valued near-field point spectrum, $E_s(\omega)$. We normalized the obtained point spectra to a reference spectrum $E_{s,Au}$, recorded on gold. A complex-valued division $(|E_s/E_{s,Au}|)$ yields the normalized near-field point spectra of the sample.

The acquisition time of an individual interferogram was 10 min. The length of one interferogram is $2 \cdot d_{max} = 800$ μm, resulting in a spectral resolution of 12 cm$^{-1}$. To obtain the spectral line scan shown in Fig. 2d, we recorded 34 point spectra along the antenna axis, with a smaller step size at the center (40 nm) than at the extremities (80 nm) of the antenna.

**Numerical simulations of s-SNOM images and spectra.** We performed the numerical calculations shown in Figs 2 and 4 using a finite element method (COMSOL). The tip was modelled by a vertically oriented point dipole source. Such an approximation takes into account that the elongated tip in the experiment is oriented perpendicular (vertical) to the sample and is illuminated by a p-polarized light. We approximate the signal detected in the far field by the vertical component of the electric field, $E_z$, below the dipole[35].

By scanning the dipole parallel to the substrate and above the h-BN resonators at a fixed frequency and recording $E_z$ below the dipole, we simulate the near-field images. In contrast, the near-field spectra are calculated by changing the frequency and maintaining the dipole at the same position. The simulated near-field spectra are normalized to the spectra above the gold substrate, $E_z/E_{z,Au}$, analogously to the experimental data treatment.

For calculations of the spectral line scan shown in Fig. 2e, the electric field below the dipole is recorded as a function of both frequency and position of the dipole.

In all simulations, the height of the dipole above the antenna was fixed to 300 nm, and the vertical component of the electric field was recorded 15 nm above the antenna's surface.

**Mode analysis.** To study the modes of the h-BN antenna (Figs 3 and 5), we performed a quasi-normal mode analysis of infinitely long h-BN waveguides. The quasi-normal mode analysis takes into account that hyperbolic phonon–polaritons in h-BN are damped and have a strong dispersion. It was performed with the COMSOL mode solver.

We considered infinitely long waveguides, which offers two advantages. First, finding the modes in an infinitely long waveguide requires only 2D simulations (instead of 3D calculations required for truncated waveguides), which computationally are much less demanding and much less time consuming. Second, analysing the modes in an infinite waveguide provides a simple and intuitive physical interpretation of the resonances in our antennas. Our approach is justified by the excellent agreement between the calculated and experimental dispersion (Fig. 3b), as well as between the calculated and experimental mode profiles shown in Fig. 5c,d, respectively.

We further assumed a rectangular cross-section for the h-BN waveguide and thus neglected the slight tilting of the fabricated antenna edges, which are seen in the Supplementary Fig. 1. It is justified for the following reasons: (i) we obtained mode dispersions and mode profiles in good agreement with the experimental results, (ii) in ref. 25 it has been shown that tilted edges have no significant influence in the optical response of hyperbolic structures and (iii) the rectangular cross-section facilitates the discussion, interpretation and understanding of the modes.

The calculations of the line profiles shown in Fig. 5b were made at frequencies slightly different from those corresponding to the experimental line scans in Fig. 5c (the frequencies are indicated in the figure caption) to match the experiment and theory due to the uncertainty in the width and height of the antenna, which strongly affects the profile of the mode.

**Antenna dimensions.** In the simulations shown in Figs 2 and 3, the length and width of the antenna were extracted from the scanning electron microscopy image shown in the Supplementary Fig. 1, while the height of the antenna was obtained from the AFM topography image shown in Fig. 2c. In the simulations shown in Figs 4 and 5, length, width and height of the antenna were obtained from AFM measurements. Notice that the width and length were taken as an average between the minimum and maximum values extracted from the AFM profile of the antenna.

**Permittivity of h-BN.** The perpendicular and parallel components of the permittivity tensor, $\epsilon_{zz} = \epsilon_\parallel$, $\epsilon_{xx} = \epsilon_{yy} = \epsilon_\perp$ are approximated with a Drude–Lorentz model[21]

$$\epsilon_a(\omega) = \epsilon_{a,\infty}\left(1 + \frac{\left(\omega_{LO}^a\right)^2 - \left(\omega_{TO}^a\right)^2}{\left(\omega_{TO}^a\right)^2 - \omega^2 - i\omega\gamma^a}\right) \quad (1)$$

Where $a = \parallel$ or $\perp$, $\omega_{LO}$ and $\omega_{TO}$ refers to the transversal (TO) and longitudinal (LO) phonon frequencies, $\gamma$ denotes the damping constant and $\epsilon_\infty$ is the high frequency permittivity. The values of the constants are: $\epsilon_{\parallel,\infty} = 2.95$, $\epsilon_{\perp,\infty} = 4.90$, $\omega_{LO}^\parallel = 825$ cm$^{-1}$, $\omega_{LO}^\perp = 1,614$ cm$^{-1}$, $\omega_{TO}^\parallel = 760$ cm$^{-1}$, $\omega_{TO}^\perp = 1,360$ cm$^{-1}$, $\gamma^\parallel = 2$ cm$^{-1}$, $\gamma^\perp = 7$ cm$^{-1}$.

**Data availability.** The data that support the findings of this study are available from the corresponding author on reasonable request.

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

## Acknowledgements

The authors acknowledge support from the European Commission under the Graphene Flagship (GrapheneCore1, Grant no. 696656), the Marie Sklodowska-Curie individual fellowship (SGPCM-705960), the ERC starting grants SPINTROS (Grant no. 257654) and 2DNANOPTICA (Grant no. 715496), the Spanish Ministry of Economy and Competitiveness (national projects FIS2014-60195-JIN, MAT2014-53432-C5-4-R, MAT2015-65525-R, MAT2012-37638 and MAT2015-65159-R), the Basque government (PhD fellowship PRE-2016-1-0150), and the Regional Council of Gipuzkoa (Project No. 100/16).

## Author contributions

A.Y.N., P.A.-G., F.J.A.-M. and R.H. conceived the study. S.V. and I.D. fabricated the h-BN antennas. F.C. and L.E.H. coordinated the fabrication. F.J.A.-M., P.A.-G., S.M. and M.A. performed the experiments. F.J.A.-M. performed the simulations. F.J.A.-M., A.Y.N., P.A.-G., P.L. and R.H. analysed the data and discussed the results. F.J.A.-M., A.Y.N. and R.H. wrote the manuscript. All authors contributed to the scientific discussion and manuscript revisions.
