## [Peer Review File · Nature Communications]

Reviewers' comments:

Reviewer #1 (Remarks to the Author):

This manuscript highlights near-field investigations of hyperbolic surface modes within hexagonal boron nitride linear antennas. While the optical modes of this material, both from nanostructures (Ref. 19 and 45) and slabs (Ref 18, 20 and 44) have been reported in both the near- and far-field, the results of this work highlight a different type of polaritonic mode that has previously not been discussed. This mode is a surface-confined waveguiding mode, in contrast to the volume confined modes discussed in prior nanostructures and in thicker slabs. Overall, the results are intriguing, highlight new physics and are worthy of publication. There are a few comments I would make in an effort to make the manuscript more clear.

1. This does raise the question of surface-confined nature of this mode and its relationship to the thickness of the structure/slab. For instance, in nanostructures, resonant modes have not been reported from structures thinner than 80 nm, and higher order branches of the HPs in the slabs are not supported in such thin flakes. Certainly in the limit of 3 monolayer thick flakes (as in Ref. 18), this mode can only be surface-confined in nature. However, the dispersion of the waveguide modes described here and that of the M0 mode in the slabs are distinct. It would be interesting to have a bit clearer description of why volume-confined modes are not observed and how these surface modes and the volume modes compare. For instance in thicker antennas of the same length and width, do simulations anticipate both present?

2. For the modes, the dispersion relationships are plotted, however, it is not clearly stated if the wavevector is the in-plane or out-of-plane q . In nanostructures, the primary modes discussed are k_z in nature, while in slabs this is k_t . In these linear antennas with a propagating mode, I would anticipate this being k_t -like (also enabling direct comparison with slab modes), but I don't see anywhere that this is stated clearly. This is also consistent with the dispersion relationship presented, but it should be stated clearly for the reader.

3. The authors show results as a function of length as well. One potential benefit this allows is to plot the q vs. antenna length for the same mode to extract the propagation length. By looking at the length where the q reaches a minimum asymptote, one can infer the propagation length and thus the authors might be able to make some more concrete discussions of how realistic an hBN waveguide with surface and/or volume confined polaritons would be for nanophotonic applications.

4. In the abstract the authors state at the end that this work provides the basis for understanding linear waveguides then goes on to include topological insulators, layered superconductors and semiconductors. However, these materials are never once mentioned in the text and therefore either the authors should expand upon how this can be applied for such material systems, or remove it from the abstract.

5. The authors define a polariton in the introduction as a "quasiparticle resulting from the coupling of light and matter". However, I don't believe this to be correct. Most people have defined this as a quasiparticle of a photon with a charged particle or electromagnetic dipole. Not to nitpick, but saying it is light coupling with matter isn't precise and should be corrected.

6. on page 6 the comment "at distances larger than 100 nm to the waveguide" I assume you mean above the waveguide? This really isn't completely clear. This paragraph in general is a bit difficult to follow and took a few reads through. I would try to focus on making this paragraph more clear.

7. When discussing the launching of HPs from the corners of structures on page 6, they might want to consider referencing Nature Comm 6, 7507 (2015) as well, which shows similar results to

ref. 44 and also displays this behaviour.

Reviewer #2 (Remarks to the Author):

In manuscript titled "Nanoimaging of resonating hyperbolic polaritons in linear boron nitride antennas", the authors reported a near-field study of the Fabry-Perot resonances in linear antennas made of a hyperbolic material. The authors (assisted by infrared nanospectroscopy and nanoimaging experiments) showed that h-BN antennae have sharp resonances with large quality factors around 100, exhibiting atypical near-field patterns. The authors used some advanced experimental techniques, which make the characterization of rectangular waveguide antennas made of hexagonal boron nitride, and consequently this work, quite interesting.

However, I've got several questions about the computation method, which is used for modal analysis, which needs clarification:

- 1) Based on modal analysis of the infinite h-BN waveguides made with a commercial mode solver the authors found that there are two types of surface modes SMO0: SMO0-A (asymmetric mode) and SMO0-S (symmetric mode). Natural resonances in polaritonic antennas are highly affected by their geometrical shape and dimensions of the antenna (e.g., length), as in fact has been demonstrated by the authors through experiments and full-wave numerical simulations. Then, is it even physically conscientious to perform such a simplified modal study for the case of h-BN antenna's analysis? Perhaps a more rigorous approach (e.g., a quasi-normal modal analysis) would be more desirable, as a numerical solver is used anyway.
- 2) By analyzing the simulation and experimental results the authors claim that SMO0-S mode is dominant. However, the authors did not comment why this mode is preferable, so the author may need to comment on this. It would be also important perhaps to make it clear how other modes could be excited.
- 3) In general, the entire purpose of the modal analysis remains elusive, and in my opinion could've been better justified.

The proposed study could be published in NC provided that the authors would address the concerns discussed above.

Response to Reviewer 1:

We thank the reviewer for the positive evaluation and for highlighting the novelty and the new physics of our work. We also thank the reviewer for the valuable and detailed comments, which helped us to improve the discussion and presentation of our manuscript. We address reviewer's comments below.

We note that during the revision of the manuscript we found a mistake we made in the measuring of the antenna dimensions and we mixed up some numbers. For that reason, we did the following in the revised manuscript:

a) We repeated the analysis and calculations shown in Figures 2 and 3. Apart of small quantitative differences, we obtained the same physical results and insights. In the revised manuscript we thus replaced Figures 2 and 3 by their corrected versions. Additionally, we provide a Supplementary Figure 1 showing how we measured the antenna dimensions, and a Supplementary Figure 2 showing how we measured the wavevectors presented in Fig. 3.

b) Figures 4 and 5 are the same as in the original manuscript but we provide the correct numbers in the figure captions.

c) We added a paragraph "Antenna dimensions" to the methods section, where we explain how antenna dimensions were measured.

1) *This does raise the question of surface-confined nature of this mode and its relationship to the thickness of the structure/slab. For instance, in nanostructures, resonant modes have not been reported from structures thinner than 80 nm, and higher order branches of the HPs in the slabs are not supported in such thin flakes. Certainly in the limit of 3 monolayer thick flakes (as in Ref. 18), this mode can only be surface-confined in nature. However, the dispersion of the waveguide modes described here and that of the M0 mode in the slabs are distinct. It would be interesting to have a bit clearer description of why volume-confined modes are not observed and how these surface modes and the volume modes compare. For instance in thicker antennas of the same length and width, do simulations anticipate both present?*

Both volume and surface-confined modes can propagate in the antennas, as we show in the mode analysis in Fig. 3. In principle, all these modes can be excited by the near fields at the tip apex. However, the M0-W mode is strongly damped in the antennas analyzed in our work, and thus exhibits extremely short propagation lengths (shorter than one polariton wavelength). In thinner antennas the propagation length would increase and we might observe the M0-W mode. The anti-symmetric mode SM0-A cannot be excited when the tip scans along the longitudinal axis of the antenna (as done in Fig. 2), as the field of the SM0-A mode has a node (vanishes) along the longitudinal antenna axis.

We follow the suggestion of the reviewer and include the following paragraph on page 6 of the revised manuscript and Supplementary Figs 3 to 5:

We note that the near fields at the tip apex, in principle, can excite any of the modes found in Fig. 3. However, in the spectral line-scans of Figs 2d,e we only observe the

SM0-S mode. We explain this finding by the strong damping of the M0-W mode, which propagates less than one polariton wavelength (see Supplementary Fig. 3). This mode might be observed in the future with wider and thinner antennas, where the damping of the mode can be significantly reduced (see Supplementary Figs. 4 and 5). On the other hand, the SM0-A mode cannot be excited when the tip is scanned along the longitudinal antenna axis, where the field of the SM0-A mode is zero. This mode, however, might be excited and observed in the future by scanning the tip, for example, along one of the long antenna edges.

2) For the modes, the dispersion relationships are plotted, however, it is not clearly stated if the wavevector is the in-plane or out-of-plane q . In nanostructures, the primary modes discussed are kz in nature, while in slabs this is kt . In these linear antennas with a propagating mode, I would anticipate this being kt -like (also enabling direct comparison with slab modes), but I don't see anywhere that this is stated clearly. This is also consistent with the dispersion relationship presented, but it should be stated clearly for the reader.

We agree with the referee that the modes propagate along the longitudinal axis of the antenna and parallel to the atomic layers of h-BN slab out of which the antennas were fabricated. To clarify this issue, we added the following comment on page 4 of the revised manuscript:

For studying the dispersion $\omega(q)$ of the HP mode yielding the longitudinal Fabry-Perot resonances, we extracted from both the experimental (Fig. 2d) and simulated spectral line-scans (see Supplementary Fig. 2) the HP wavevector q_n at the resonance frequency ω_n of the longitudinal n -th order mode. **The wavevectors q_n are parallel to the longitudinal axis of the antenna (y-axis) and to the atomic layers of the h-BN slab out of which the antennas were fabricated. In good approximation, they can be calculated by using the relation $q_n = \pi/L_n$, where L_n is the distance between the adjacent maxima in the central part of the antenna (for each resonance order $n > 2$). To be more precise, we determined the q_n by fitting the line profiles with a sine function, as described in the Supplementary Fig. 2. In Fig. 3b we show the dispersion obtained from experiment (filled symbols) and simulation (open symbols). (...)**

3) The authors show results as a function of length as well. One potential benefit this allows is to plot the q vs. antenna length for the same mode to extract the propagation length. By looking at the length where the q reaches a minimum asymptote, one can infer the propagation length and thus the authors might be able to make some more concrete discussions of how realistic an hBN waveguide with surface and/or volume confined polaritons would be for nanophotonic applications.

The referee proposes an interesting analysis, which unfortunately is not possible with our current data. On the other hand, near-field microscopy would allow to directly measure the propagation length by simply scanning a h-BN rod which is significantly longer than the propagation length. Our current antennas are too short for such experiments, which are anticipated for the future.

4) In the abstract the authors state at the end that this work provides the basis for understanding linear waveguides then goes on to include topological insulators, layered superconductors and semiconductors. However, these materials are never once mentioned in

the text and therefore either the authors should expand upon how this can be applied for such material systems, or remove it from the abstract.

In the revised manuscript we removed the last phrase from the abstract:

Our work establishes the basis for the understanding and design of linear waveguides, resonators, sensors and metasurface elements based on hyperbolic materials and metamaterials., ~~including topological insulators, and layered superconductors and semiconductors.~~

5) The authors define a polariton in the introduction as a "quasiparticle resulting from the coupling of light and matter". However, I don't believe this to be correct. Most people have defined this as a quasiparticle of a photon with a charged particle or electromagnetic dipole. Not to nitpick, but saying it is light coupling with matter isn't precise and should be corrected.

We agree that our phrase is not precise enough and thus we changed it the revised manuscript (first paragraph in the first page):

Polaritons - quasiparticles resulting from the coupling of photons and oscillating charges¹ - in nanostructured materials enable strong confinement and enhancement of electromagnetic fields at extreme subwavelength-scale dimensions².

6) On page 6 the comment "at distances larger than 100 nm to the waveguide" I assume you mean above the waveguide? This really isn't completely clear. This paragraph in general is a bit difficult to follow and took a few reads through. I would try to focus on making this paragraph more clear.

The phrase "at distances larger than 100 nm to the waveguide" indeed means above the waveguide. In order to be more precise we corrected the corresponding paragraph on page 7 of the revised manuscript, trying to make the writing of the whole paragraph clearer.

Inside the h-BN waveguide, we recognize 'zig-zag' patterns. They manifest HP rays that emerge from corners^{19,44,45} and reflect at the top and bottom h-BN waveguide surfaces⁴⁶. As the propagation angle θ of the hyperbolic rays depends on the frequency ω (given by¹⁹ $\tan \theta(\omega) = i\sqrt{(\epsilon_{\perp}(\omega)/\sqrt{\epsilon_{\parallel}(\omega)})}$), the multiple reflections yield complex and frequency-dependent field distributions inside the waveguide, which extend several nanometers above the top waveguide surface. Scanning the tip in close proximity across the top waveguide surface (perpendicular to the longitudinal waveguide axis) thus leads to the transversal near-field oscillations observed in Fig. 5c (marked by green arrows). On the other side, we observe in Fig. 5a a rather homogeneous near-field distribution at distances larger than 100 nm both above the top waveguide surface and below the bottom surface. Its structure (opposite sign of the vertical component of the electric field above and below the waveguide) does not change with frequency, which reveals that the different near-field profiles shown in Figs. 5c and d belong to the same waveguide mode.

7) When discussing the launching of HPs from the corners of structures on page 6, they might want to consider referencing Nature Comm 6, 7507 (2015) as well, which shows similar results to ref. 44 and also displays this behavior.

In the revised manuscript we included the corresponding reference:

They manifest HP rays that emerge from corners^{19,44,45} and reflect at the top and bottom h-BN waveguide surfaces⁴⁶.

45. Li, P. et al. Hyperbolic phonon-polaritons in boron nitride for near-field optical imaging and focusing. Nat. Commun. 6, 7507 (2015).

Response to Reviewer 2:

We appreciate the positive comments and suggestions of the reviewer, which help us to improve the discussion of the theoretical aspects of our manuscript.

We note that during the revision of the manuscript we found a mistake we made in the measuring of the antenna dimensions and we mixed up some numbers. For that reason, we did the following in the revised manuscript:

- a) We repeated the analysis and calculations shown in Figures 2 and 3. Apart of small quantitative differences, we obtained the same physical results and insights. In the revised manuscript we thus replaced Figures 2 and 3 by their corrected versions. Additionally, we provide a Supplementary Figure 1 showing how we measured the antenna dimensions, and a Supplementary Figure 2 showing how we measured the wavevectors presented in Fig. 3.
- b) Figures 4 and 5 are the same as in the original manuscript but we provide the correct numbers in the figure captions.
- c) We added a paragraph “Antenna dimensions” to the methods section, where we explain how antenna dimensions were measured.

We answer point 1 and 3 of the referee together, as they are closely related:

1) Based on modal analysis of the infinite h-BN waveguides made with a commercial mode solver the authors found that there are two types of surface modes SMO0: SMO0-A (asymmetric mode) and SMO0-S (symmetric mode). Natural resonances in polaritonic antennas are highly affected by their geometrical shape and dimensions of the antenna (e.g., length), as in fact has been demonstrated by the authors through experiments and full-wave numerical simulations. Then, is it even physically conscientious to perform such a simplified modal study for the case of h-BN antenna's analysis? Perhaps a more rigorous approach (e.g., a quasi-normal modal analysis) would be more desirable, as a numerical solver is used anyway.

3) In general, the entire purpose of the modal analysis remains elusive, and in my opinion could've been better justified.

We understand quasi-normal modes as modes of a lossy system. The absorption (the imaginary part of ϵ) is taken into account in our full-wave simulations of the modes in the infinite waveguide. These modes propagate in space and have complex-valued wave-vectors at a given (real) frequency. In the revised manuscript we describe in the “*Mode analysis (Figures 3 and 5)*” subsection of the *Methods* section that we perform the mode analysis in a lossy system.

On the other hand, the referee refers to the finite length of the antenna, which indeed has not been taken into account in our mode analysis. We did not perform the quasi-normal mode analysis in the finite-length antennas (truncated waveguides). We considered the modes of the simplified system (the infinite waveguide) for the following two main reasons: First, finding the modes in the infinite waveguides requires 2D simulations (instead of 3D ones in the case

of truncated waveguides), which computationally are significantly less demanding and time-consuming. Second, by analyzing the modes in the infinite waveguide we gain a simple and intuitive physical interpretation of the resonances in our antennas. For example, we analyzed the asymptote of the waveguide modes to classify them as volume or surface hyperbolic polaritons. Additionally, we compared the experimental near-field profiles with the calculated field profiles of the SM0-S mode in Fig. 5, finding an excellent agreement between both of them. In general, for our linear antennas the use of the basis of the quasi-normal modes of the infinite waveguide is justified since we can approximate the antenna as a cavity in which the waveguide modes reflect forth and back, creating Fabry-Perot resonances.

In the revised version of the manuscript we improved the justification of the mode analysis in the abstract, conclusions and main text.

Abstract:

By performing a detailed mode analysis we can assign the antenna resonances to a single waveguide mode originating from the hybridization of hyperbolic surface phonon polaritons (Dyakonov polaritons) that propagate along the edges of the h-BN waveguide.

Main Text, Page 5:

Both findings indicate that highly confined waveguide polaritons are the cause of the longitudinal Fabry-Perot resonances. To identify the waveguide mode, we performed a numerical analysis of the electromagnetic modes in an infinitely long h-BN waveguide, which has a cross-section that is similar to that of the antenna (see ‘Methods’ section).

Discussion, Page 7:

By a detailed mode analysis we demonstrated that the waveguide mode exhibiting Fabry-Perot resonances originates from the hybridization of hyperbolic surface phonon polaritons (Dyakonov polaritons) that propagate along the edges of the h-BN antennas (respectively the h-BN waveguides).

Mode analysis (Figures 3 and 5), Methods:

To study the modes of the h-BN antenna (Fig. 3 and 5), we performed a quasi-normal mode analysis of infinitely long h-BN waveguides. The quasi-normal mode analysis takes into account that hyperbolic phonon polaritons in h-BN are damped and have a strong dispersion. It was performed with the COMSOL mode solver.

We considered infinitely long waveguides, which offers two advantages. First, finding the modes in an infinitely long waveguide requires only 2D simulations (instead of 3D calculations required for in truncated waveguides), which computationally are much less demanding and much less time-consuming. Second, analyzing the modes in an infinite waveguide provides a simple and intuitive physical interpretation of the resonances in our antennas. Our approach is justified by the excellent agreement between the calculated and experimental dispersion (Fig. 3b), as well as between the calculated and experimental mode profiles shown in Fig. 5c and d, respectively.

We further assumed a rectangular cross-section for the h-BN waveguide and thus neglected the slight tilting of the fabricated antenna edges, which are seen in the Supplementary Fig. 1. It is justified for the following reasons: (i) we obtained mode

dispersions and mode profiles in good agreement with the experimental results, (ii) in Ref. 23 it has been shown that tilted edges have no significant influence in the optical response of hyperbolic structures, and (iii) the rectangular cross section facilitates the discussion, interpretation and understanding of the modes.

2) *By analyzing the simulation and experimental results the authors claim that SM0-S mode is dominant. However, the authors did not comment why this mode is preferable, so the author may need to comment on this. It would be also important perhaps to make it clear how other modes could be excited.*

Both volume and surface-confined modes can propagate in the antennas, as we show in the mode analysis in Fig. 3. In principle, all these modes can be excited by the near fields at the tip apex. However, the M0-W mode is strongly damped in the antennas analyzed in our work, and thus exhibits extremely short propagation lengths (shorter than one polariton wavelength). In thinner antennas the propagation length would increase and we might observe the M0-W mode. The anti-symmetric mode SM0-A cannot be excited when the tip scans along the longitudinal axis of the antenna (as done in Fig. 2), as the field of the SM0-A mode has a node (vanishes) along the longitudinal antenna axis.

We follow the suggestion of the reviewer and include the following paragraph on page 6 of the revised manuscript and Supplementary Figs 3 to 5:

We note that the near fields at the tip apex, in principle, can excite any of the modes found in Fig. 3. However, in the spectral line-scans of Figs 2d,e we only observe the SM0-S mode. We explain this finding by the strong damping of the M0-W mode, which propagates less than one polariton wavelength (see Supplementary Fig. 3). This mode might be observed in the future with wider and thinner antennas, where the damping of the mode can be significantly reduced (see Supplementary Figs. 4 and 5). On the other hand, the SM0-A mode cannot be excited when the tip is scanned along the longitudinal antenna axis, where the field of the SM0-A mode is zero. This mode, however, might be excited and observed in the future by scanning the tip, for example, along one of the long antenna edges.

REVIEWERS' COMMENTS:

Reviewer #1 (Remarks to the Author):

Overall I am satisfied with the changes the authors have made and believe the paper is ready for publication. I had a couple final comments that the authors might want to consider, but don't believe they are required to be included for publication. There are a couple typos that need to be addressed:

1) page 3: "We first perform nano-FTIR spectroscopy (Fig. 2a) of a representative, 1.723 nm long h-BN rod antenna" I assume that should be um not nm?

suggested comments:

1) I assume these are isolated antennas and therefore far-field spectra weren't possible? In Giles et al., Nano Lett. 16(6), 3858 (2016) it was shown that differences between the near- and far-field spectra were limited only to spectral shifts from the presence of the tip. Presuming that is also likely the case here, it may be useful to explain that while the tip is providing a scattering source that all modes observed are predicted by the 2D modeling and thus, all modes are expected to be present with only far-field excitation. This would be beneficial as in most potential uses only those modes that can be stimulated via external excitation and/or observed in the far-field would be of significant use.

2)

Reviewer #2 (Remarks to the Author):

Photonic antennas made of hyperbolic materials could have strong application potential for IR nanophotonics. The most canonical type of antennas – linear waveguide antennas made of the van der Waals material h-BN has been studied. According to the full-wave simulations, a comprehensive analysis of both the experimental images and near-field spectra is performed. The authors show that the waveguide mode exhibits Fabry-Perot resonances and originates from the hybridization of hyperbolic surface phonon polaritons that propagate along the edges of the h-BN antennas. The manuscript and the supplementary materials are now presented sufficiently well, and the experimental details are described more clearly. In my opinion, the new version of the manuscript is now ready for publication.

Response to Reviewer 1:

We would like to thank the referee again for his positive comments about the manuscript and for its insightful remarks.

1) page 3: "We first perform nano-FTIR spectroscopy (Fig. 2a) of a representative, 1.723 nm long h-BN rod antenna" I assume that should be um not nm?

We corrected this typo in the revised version, page 3:

We first perform nano-FTIR spectroscopy (Fig. 2a) of a representative, 1723 nm long h-BN rod antenna

2) I had a couple final comments that the authors might want to consider, but don't believe they are required to be included for publication.

I assume these are isolated antennas and therefore far-field spectra weren't possible? In Giles et al., Nano Lett. 16(6), 3858 (2016) it was shown that differences between the near- and far-field spectra were limited only to spectral shifts from the presence of the tip.

The referee is right when he points out that far-field experiments were not possible. Indeed, antennas with different sizes were fabricated out of the same h-BN flake, making this sample not suitable for far-field experiments, where usually numerous antennas of identical dimensions are illuminated. Due to the lack of far-field experiments a detailed comparison between near-field and far-field spectra of the h-BN nanorod antennas is out of the scope of this article.

Presuming that is also likely the case here, it may be useful to explain that while the tip is providing a scattering source that all modes observed are predicted by the 2D modeling and thus, all modes are expected to be present with only far-field excitation. This would be beneficial as in most potential uses only those modes that can be stimulated via external excitation and/or observed in the far-field would be of significant use.

This is a good point. We thus added the following comment (marked in red colour) to the discussion on Page 7,8:

Page 7,8, Discussion:

In conclusion, we employed real-space infrared nanoimaging and nanospectroscopy for studying Fabry-Perot resonances of hyperbolic phonon polaritons in finite-length h-BN waveguides acting as infrared antennas. We found resonances exhibiting Q -factors up to 120, which make the antennas interesting building blocks for the development of infrared molecular sensors, narrowband thermal emitters or metasurfaces for flat infrared photonic elements. By a detailed mode analysis we demonstrated that the waveguide mode exhibiting longitudinal Fabry-Perot resonances originates from the hybridization of hyperbolic surface phonon polaritons (Dyakonov polaritons) that propagate along the edges of the h-BN antennas (respectively the h-BN waveguides). This mode exhibits a stronger field confinement (i.e. larger wavevector) compared to the waveguides' volume

modes (of the same order). **We note that the odd longitudinal Fabry-Perot resonances could be also excited by far-field illumination, which could enable, for example, field-enhanced far-field spectroscopy applications.** From a general perspective, our results provide valuable insights into the fundamental properties of polariton modes in deep subwavelength-scale linear waveguides based on naturally or artificially layered materials, such as van der Waals materials and heterostructures or metal-dielectric metamaterials. The knowledge about the mode properties will be of critical importance for the development of photonic circuits based on hyperbolic plasmon-polariton and phonon-polaritons in linear waveguides and antennas.